# An AI-Based Colonic Polyp Classifier for Colorectal Cancer Screening Using Low-Dose Abdominal CT

**DOI:** 10.3390/s22249761

**Published:** 2022-12-13

**Authors:** Islam Alkabbany, Asem M. Ali, Mostafa Mohamed, Salwa M. Elshazly, Aly Farag

**Affiliations:** 1Computer Vision and Image Processing Laboratory, University of Louisville, Louisville, KY 40292, USA; 2Kentucky Imaging Technologies, LLC, Louisville, KY 40245, USA

**Keywords:** CTC, FI, AI, polyp detection

## Abstract

Among the non-invasive Colorectal cancer (CRC) screening approaches, Computed Tomography Colonography (CTC) and Virtual Colonoscopy (VC), are much more accurate. This work proposes an AI-based polyp detection framework for virtual colonoscopy (VC). Two main steps are addressed in this work: automatic segmentation to isolate the colon region from its background, and automatic polyp detection. Moreover, we evaluate the performance of the proposed framework on low-dose Computed Tomography (CT) scans. We build on our visualization approach, Fly-In (FI), which provides “filet”-like projections of the internal surface of the colon. The performance of the Fly-In approach confirms its ability with helping gastroenterologists, and it holds a great promise for combating CRC. In this work, these 2D projections of FI are fused with the 3D colon representation to generate new synthetic images. The synthetic images are used to train a RetinaNet model to detect polyps. The trained model has a 94% f1-score and 97% sensitivity. Furthermore, we study the effect of dose variation in CT scans on the performance of the the FI approach in polyp visualization. A simulation platform is developed for CTC visualization using FI, for regular CTC and low-dose CTC. This is accomplished using a novel AI restoration algorithm that enhances the Low-Dose CT images so that a 3D colon can be successfully reconstructed and visualized using the FI approach. Three senior board-certified radiologists evaluated the framework for the peak voltages of 30 KV, and the average relative sensitivities of the platform were 92%, whereas the 60 KV peak voltage produced average relative sensitivities of 99.5%.

## 1. Introduction

Colorectal cancer (CRC) originates as small growths (polyps) attached to the luminal wall of the colon and rectum. If the polyps are not diagnosed and treated, they may grow in size and become cancerous. Untreated colorectal cancer spreads from the local invasion of the colon and rectum (in situ) into surrounding tissues and lymph nodes (regional), and eventually to distant parts of the body, such as the liver and lungs. Colorectal cancer is the third most common cancer in the US, and it is also the second leading cause of cancer deaths, behind lung cancer. The American Cancer Society (ACS) recommends that people that are at average risk of colorectal cancer should start regular screening at age 45. The ACS also has recommendations for higher age groups, based on health, up to 85 [1]. The US population of between 45–75 years of age (recommended for colonoscopy) is nearly 52.2 M, according to the 2021 consensus. Although the statistics for US Veterans are not precisely known, over 4000 cases of colorectal cancer are diagnosed each year among Veterans [2,3], and it is estimated that 1.5 million colonoscopies are required at present for them. Therefore, accurate CTC at a reduced X-ray dosage will benefit the entire population recommended for colonoscopy; it will especially benefit rural and economically depressed regions in the US.

There are four common methods used in screening for CRC: (1) Fecal occult blood test (FOBT), which detects blood that is not visible in a stool sample; (2) a fecal immunochemical test (FIT) that detects occult blood in stools (note: FIT, multitarget stool DNA (mt-sDNA)/Cologuard, and FOBT are “stool-based test” looking for occult blood); (3) Optical Colonoscopy (OC), where a flexible endoscope is inserted to visually inspect the interior walls of the rectum and colon; and (4) Computed Tomography Colonography (CTC), a CT-based technology for visualizing the interior of the colon. Pickhardt et al. 2020 [4] introduced an extensive survey on the diagnostic performance of the available noninvasive CRC screening tests, including mt-sDNA testing, fecal immunochemical testing (FIT), and CT Colonography (CTC), with an emphasis on the comparison of positive predictive values (PPV) and detection rate (DR) for advanced neoplasia (AN; encompassing cases of advanced adenomas and CRC), showed that among all of the noninvasive CRC screening tests, CTC with a polyp size threshold of 10 mm or larger most effectively targets AN, preserving detection while also decreasing unnecessary colonoscopies compared with mt-sDNA testing and FIT. Among the non-invasive CRC screening approaches, CTC is much more accurate in terms of sensitivity and specificity for polyp detection [4].

The American College of Radiology Imaging Network (ACRIN) [5] performed standard CTC and OC on 2531 patients at 15 study centers in the US. Using OC as the ground-truth (gold standard), the study showed that per-patient (i.e., patient’s own CTC vs. OC), sensitivity, specificity, positive and negative predictive values (PPV and NPV), and area under the receiver-operating-characteristic curve (ROC) for CTC were: 0.9±0.03, 0.86±0.02, 0.23±0.02, 0.99±0.01, and 0.89±0.02, respectively. The per-polyp sensitivity for large adenomas or cancers was 0.84±0.04, and the per-patient sensitivity for detecting adenomas that were ≥6 mm in diameter was 0.78. These figures are very close to OC. Various other studies, e.g., [6,7,8], concluded the same sensitivity and specificity of CTC vs. OC.

The cost effectiveness of CTC vs. OC was investigated by Pyenson and Pickhardt et al. [9]. Their study looked at the Medicare cost, specifically for CTC versus colonoscopy. Based on the calculations and models utilized in this study, CTC appears to be 29% less expensive than traditional colonoscopy. The researchers analyzed various scenarios that may change the costs of either CTC or colonoscopy; for example, the number of screenings for CTC versus colonoscopy. In an analysis of nine additional scenarios and the combinations of said scenarios, CTC always came out ahead from a cost standpoint, with a range of cost advantage ranging from 12% to 58%.

### 1.1. Background

From a computational perspective, the Tomography Colonography (CTC) pipeline involves five steps (Figure 1): (1) image processing (e.g., electronic colon cleansing) to correct prep and scanner errors; (2) colon segmentation to isolate the colon tissue from the rest of the abdomen; (3) 3D reconstruction to generate a volumetric representation of the colon; (4) luminal surface visualization; and (5) analysis to catalog the detected polyp in terms of location, size, shape, and potential pathology.

CTC Visualization: A Computed Tomography Colonography (CTC) visualization methodology is used to visualize the colon luminal surface. There are different CTC visualization approaches (e.g., [10,11,12,13,14,15,16,17,18,19,20]) that have been developed in the literature.

Fly-Through (FT) [10,11] mimics Optical Colonoscopy (OC). A virtual camera moves along the centerline/medial axis of a colon to render its internal views. Up to 20% of colonic polyps can be missed using unidirectional FT. Therefore, to examine the entire colon, FT is performed in the antegrade and retrograde directions.

Fly-Over (FO) [18] slices the colon into two halves along the centerline/medial axis, and uses two virtual cameras for visualization; one for each half. From an imaging perspective, FO has an adaptable FOV, enabling a better visualization capability than other visualization methods (i.e., it provides better sensitivity and specificity for polyp detection). FO is computationally challenging: the cutting/slicing plane may include polyps, and “holes” can result in the surface and result in missed opportunities for polyp detection.

Fly-In (FI) [20] performs FO internally, using a ring of virtual cameras around the centerline, showing a 360° field of view (FOV) of the colon segments internally. FI resolves several issues hampering the implementation of FO while providing a longitudinal visualization (filet-visualization) without distortion. Thus, the established advantage of FO vs. FT is maintained, while creating a distortionless filet that is far superior to Colon Flattening. By moving the ROI along the centerline of the colon, radiologists would be able to examine the luminal surface and detect colonic polyps.

### 1.2. Related Work

Several studies in the literature have worked on automatic polyp detection in optical colonoscopy and fly through virtual colonoscopy. Nadeem et al. [21] used a machine learning algorithm to obtain a depth map for a given optical colonoscopy image, and then they used a detailed pre-built polyp profile to detect and to delineate the boundaries of polyps in the given image. They used images generated from fly-through virtual colonoscopy for the training and real optical colonoscopy images for testing.

Livovsky et al. [22] developed a polyp detection system based on deep learning that alerts the operator in real time to the presence and location of polyps. They evaluated the system performance on a large OC video set containing 1393 procedures. They analyzed false positives and found a subgroup of ‘subtle polyps’ that have been missed by endoscopists.

Pickhardt et al. [23] made a systematic review and analysis of the published CT Colonography (CTC) studies published between January 1994 and August 2017, and assessed polyp detection in senior-age (≥65 years old) cohorts. They included 34 studies with 41,680 (18,798 senior-age), and found that on average that one in every 12 senior-age adults without symptoms of CRC who underwent screening would be referred to colonoscopy using the 10 mm threshold, with a high yield for advanced neoplasia and a high sensitivity for cancer detection.

Table 1 compares between different approaches that are used for polyp detection. It illustrates the strengths and weakness of each categories. Our goal in this work is to develop a non-invasive automatic polyp detection approach using virtual colonoscopy. Unlike optical colonoscopy, the proposed approach can be used in a massive data screening. In addition, it helps radiologists to accurately read abdominal scans in short time by detecting the polyp candidates. This proposed approach could benefit many clinics, especially the ones in rural and economically depressed regions that have a limited number of gastroenterologists to perform optical colonoscopy, and experienced radiologists to read the scans.

Table 2 presents some examples of the research that has been performed for automatic polyp detection. As shown, many studies have good detection rates; however, they use optical colonoscopy video frames to detect the polyps, which is an invasive method.

Approaches using deep learning can classify a massive number of objects in natural and medical images (e.g., [22,24,25]). Our group [26] showed that curvature-based methodologies yield a significant degree of accuracy for the detection of sessile polyps.

Our hypothesis is that a proper synchronization of CTC and OC holds the greatest promise for combating CRC by enhancing population uptake and increasing screening. For 20 years, our research team has worked on building a front-end CTC visualization system. We contributed to each step of the CTC pipeline: preprocessing  [31], centerline extraction [32], and visualization [19,33,34].

In this work, we mainly focus on the last step of the CTC pipeline. The main contributions of this work are:A new automatic colon segmentation technique that succeeds in 90% of scans to automatically segment the colon region from CT.A new approach for automatic polyp detection from the Fly-In projections.Evaluation of the performance of the proposed framework on low-dose CT scans.

## 2. Proposed Approach

### 2.1. Colon Segmentation

Colon segmentation is a challenging problem because the colon has asymmetric topology. In addition, uncertainties appear due to the presence of Hounsfield intensity regions consisting of air, soft tissue, and high-attenuation structures similar to bone. In addition, complications result due to the presence of residual stool, lesions, and disconnected colon segments.

The proposed segmentation approach involves multiple steps, as shown in Figure 2. The first step is to detect the probable air regions in the DICOM image, based on thresholding. The second step is to collect the segmented potential air regions as connected components. The third step uses the Markov Random Field (MRF)-based algorithm to determine which segment could belong to the colon. Then, we detect the high-intensity (fluid) regions below the air regions inside the colon by region growing toward the direction of gravity in the DICOM’s high intensity threshold.

The problem is formulated as a Maximum-A-Posterior (MAP) estimation of a Markov Random Field (MRF). To segment a volume, we initially labeled the volume based on its gray level probabilistic model. Then, we create a weighted undirected graph with vertices corresponding to the set of volume voxels P, and a set of edges connecting these vertices. Consider a neighborhood system N of all unordered pairs {i;j} of neighboring voxels in P. Let L be the set of labels corresponding to the colon and its background, and denote the set of labeling by f. The goal is to find the optimal segmentation; best labeling *f*, by minimizing the function:(1)$$E\left(f\right)=\sum_{\left\{i,j\right\}}^{N}V\left(f_{i},f_{j}\right)+\sum_{i\in P}D\left(f_{i}\right),$$
where $D(f_{i})$ measures how much assigning a label $f_{i}$ to voxel *i* disagrees with the voxel intensity $I_{i}$. The second term is the pairwise interaction model that represents the penalty for the discontinuity between voxels *i* and *j*. Optimization is carried out using the Graph Cut approach [35].

After a colon is segmented, 3D reconstruction used to generate a volumetric representation of the colon is performed. Then, a visualization methodology is used to visualize the colon luminal surface. Figure 1 shows the colon representation for many of these visualization methods.

The Fly-In approach provides state-of-the-art performance in terms of sensitivity and specificity for the detection of colonic polyps ≥6 mm [36]. Therefore, we build upon the FI approach for polyp detection. In the proposed framework, FI is used to generate 2D images for the internal surface of the colon. These 2D images are fused with 3D surface information. Then, a RetinaNet model is trained using these fused images to detect polyps.

Optimal visualization projects most of the surface cells on the image plane without local deformations or loss. The visualization loss can be defined as a function of three factors: The angle (α) between the projection direction (*p*) and the camera’s principal axis ($\vec{look}$), the angle (Φ) between the projection direction (*p*) and the cell’s normal vector (*n*), and finally, the ratio between the camera focal length (*f*) and the cell’s distance (*d*) to the projection center in the direction of ($\vec{look}$). Figure 3 demonstrates the FI on a rendered segment of the colon and shows a rig of eight cameras over a ring. A near-distortionless filet of the colon segment is shown as well. By adjusting the visualization frustum for the cameras, we can control the size of the displayed ring and its resolution. Figure 4 shows the results using the actual CTC data of 12 patients visualized via the FT, FO, and FI methods, normalized using the function $F_{L_{v}}$. The results show that Fly-In approach provides state-of-the-art performance in visualization, which is promising for polyp detection.

### 2.2. Polyp Detection

Our hypothesis is that fusing the 2D projections and the 3D colon representation in virtual colonoscopy may enhance polyp detection accuracy. Virtual cameras in the FI approach generate a “filet”-like 2D image of the internal surface of a colon ring. This projection can be visualized as an RGB image (Figure 5a). To add the 3D information to an RGB image, we calculate the curvature at each point on the surface, then the curvature value is represented as an RGB value, as shown in Figure 5b.

Our proposed polyp detection should be fast enough to run in real time; therefore, when it came to choosing between one-stage and two-stage detectors, we preferred a one-stage detector as it runs faster. However, a one-stage detector such as YOLO (You Only Look Once) [37,38] and SSD (Single Shot Detector) [39] are fast detectors, they do not achieve a better accuracy than the two-stage detectors due to a significant imbalance between the object and background classes. The polyp detection problem is an example of training a model using an imbalanced dataset. Therefore, we used the RetinaNet model [40] for polyp detection (see Figure 6). RetinaNet [40] is a one-stage object detection model that utilizes a focal loss function to address class imbalance during training. Focal loss balance between easy and hard examples particularly downweighs the easy examples and focuses on the hard examples. The focal loss also balances between the positive and negative examples by using a weighting parameter, which is usually the inverse class frequency to penalize the model.

The focal loss is defined as follows:(2)$$FL=-{\left(1-p_{t}\right)}^{\gamma }log\left(p_{t}\right)$$
where γ≥0 is a hyper-parameter which allows us to adjust the rate at which easy examples are down-weighted smoothly, and $p_{t}$ is the estimated probability of ground truth. $p_{t}$ gives the probability of class 1 when the ground truth is 1 and the probability of class 0 when the ground truth is 0.
(3)$$p_{t}=\left\{\begin{array}{cc} p & \;y=1\\ \left(1-p\right) & \;otherwise \end{array}\right.$$
where *y* is the ground truth class ( 1 for polyp and 0 for non-polyp).

Retinanet uses a Feature Pyramid Network (FPN) that provides a rich multi-scale feature pyramid by implementing a top-down approach with lateral connections. Therefore, it can detect polyps with various sizes. Using these two techniques (Focal loss and Feature Pyramid Network), RetinaNet [40] detectors outperform all previous one-stage and two-stage detectors. In addition, RetinaNet [40] detectors are faster than other detectors with a similar accuracy. For the training process, the Adam optimizer with a learning rate of $10^{-5}$ is used to minimize the detection loss. The learning rate is automatically reduced when a metric has stopped improving. Also, we use gradient clipping to avoid the gradient exploding problem.

## 3. Low-Dose CTC Visualization Using the FI Approach

A CT scan allows for the non-invasive acquisition of the organs of the human body. This acquisition is based on the fact that X-ray beams have different attenuations depending on the medium (air, water, and tissues), which results in an internal distribution of the body organs being obtained. Analytical techniques such as filtered back-projection (FBP) or iterative reconstruction (IR) are used for this task. These techniques could be easily applied in the standard condition (a high dose that results in a high signal-to-noise ratio). However, high doses of radiation could be potentially harmful. Therefore, developing an accurate FI-based polyp visualization from low-dose CT scans will be significant for the recurrent CTC scans that are currently recommended by the American College of Radiology.

Modern scanners focus on reducing the radiation dose, which results in more challenges for the reconstruction algorithm because it causes an increase in the noise levels, which reduces the signal-to-noise ratio. While traditional analytical methods may fail, this task does not hold prior knowledge regarding the target reconstructions. So, adding machine learning approaches helps the model to learn underlying distributions.

### 3.1. Generating Low-Dose CTs from Existing Retrospective CT Scans

To generate synthetic scans with different low doses, i.e., low-dose CT (LDCT) scans from the available retrospective CT scans, we should simulate the reduction in the tube peak voltages from the standard CT tube peak voltage of 120 kV. Various approaches exist for generating low-dose image slices from a higher-dose image. (i) Body CT: Yu et al. [41] developed a noise insertion method to simulate lower-dose images from the existing standard-dose computed tomography (CT) data. In their noise insertion method, they integrated the effects of the bowtie filter, automatic exposure control, and electronic noise. (ii) LIDC/IDRI database: Leuschner et al. [42] created a database of computed tomography images and simulated low photon count measurements. They used slices from the LIDC/IDRI Lung database. (iii) Phantoms: Takenaga et al., [43] simulated low-dose CT by adding noise to a high-dose CT image reconstructed using a filtered back-projection algorithm. They used an ACR phantom for validation. Elhamiasl and Nuyts [44] modeled X-ray tube current reduction by estimating the noise-equivalent number of photons in the high exposure scan, then they applied a thinning technique to reduce that number. Zeng et al. [45] simulated low-dose CT transmission data by adding an independent Poisson noise distribution plus independent Gaussian noise distribution.

In this work, a similar approach is used to generate Low-Dose CT (LDCT) from the CTC data by injecting Poisson and Gaussian noises. The Poisson noise models Quantum noise that stems from the process of photon generation, attenuation, and detection, while Gaussian noise models the detector noise that stems from the electronic data acquisition system. A detailed method is described in Algorithm 1.
**Algorithm 1:** Generating LDCT from existing retrospective CT scans1:HU: Dequantise Hounsfeld unit (HU) values of original DICOM scan by adding uniform noise [0, 1)2:μ: Compute the linear attenuations values $\mu =HUx0.001x\left(\mu_{water}-\mu_{air}\right)+\mu_{water}$ using μ values of the deiserd dose tube voltage, then normalize values in range [0, 1]3:*R*: Project the data using the ray transform (Radon transform in 2D) for CT4:*T*:Convert the high-dose sinogram data to the transmission one $T=e^{\left(-R\right)}$5:$T_{LD}$: Generate the simulated low-dose transmission data by injecting Poisson and Gaussian noise. The Poisson noise models the quantum noise that stems from the process of photon generation, attenuation and detection, while Gaussian noise models the detector noise that stems from the electronic data acquisition system6:$R_{LD}$: Convert the low-dose transmission data to sinogram data $R_{LD}=-ln\left(T_{LD}\right)$7:$\mu_{LD}$: Ray Data back Projection to obtain linear attenuations μ for the low-dose CT8:$HU_{LD}$: Compute the new HU value for the low-dose CT $HU_{LD}=1000x\left(\mu -\mu_{water}\right)/\left(\mu_{water}-\mu_{air}\right)$

### 3.2. Evaluating CTC at a Low Dose

To evaluate FI on low-dose CT and to simulate virtual CTC, a deep learning approach (FBP + U-Net [46,47]) is used to enhance the low-dose CT. FBP [46] is a widely used analytical CT reconstruction technique, but the resulting CT includes a lot of artefacts. Therefore, a second stage of denoising is required. For the FBP [46], we used the Hann filter with a frequency scaling of 0.641. For the post-processing approach (FBP+U-Net [46,47]), we used a U-Net-like [47] architecture with five scales to enhance the low-dose CT images. Figure 7(i) shows a summary of this approach.

We simulate the reduction in the tube peak voltages using CT, starting from the standard tube peak voltages of 120 kV, to obtain CTs with different dose levels (80, 60, and 30 KV). A U-Net model is trained on the LoDoPaB-CT Dataset [42] to obtain the reconstructed enhanced CT. Figure 7(ii) shows an example of low-dose and enhanced CT images using the proposed algorithm. As shown in Figure 7, the FI visualization of the enhanced low-dose CTC is very close to the original standard-dose CTC visualization, while the FI visualization of the low-dose CT misses the polyp.

## 4. Experiments

**Data:** Two sets of data are used to validate our proposed framework. For segmentation evaluation, we use a dataset provided by the American College of Radiology Imaging Network (ACRIN) [5] and the Walter-Reed medical center. This dataset contains 254 DICOM scans that were manually annotated by experts who provided a segmentation mask of the colon. Another dataset is used for training and validating the polyp detection module; this was provided by Dr. Pickhardt from University of Wisconsin hospitals. The data consists of scans in the supine and prone positions. These scans are for 49 patients of ages [40–77 years] (23 males 23 females, and 3 unknown genders), and the scans have 59 annotated polyps that are higher than 6 mm. A CT scanner using a low-dose protocol, with collimation 128 × 0.6 mm, slice thickness 1.0 mm, a reconstruction interval 0.7 mm, and a tube peak voltage KVP 120, with one case having a 140 tube current [29–400 mA], a mean of 172.33, and a std of 80.36; exposure times of [446–623 ms], a mean of 468 and a std of 39.55; and a slice thickness of 1.25 mm, was captured in all cases. Lesion detection was performed by one of three experienced radiologists. Images were read in primary 2D (window setting 1500, ±250 HU).

**Segmentation:** To evaluate our automatic segmentation approach, we used the first dataset of 254 scans. Then, we compared the automatic segmentation S to the ground truth G using the intersection over the union metric, $IoU=\frac{G\cap S}{G\cup S}$. The obtained results are shown in Figure 8. The results show that our automatic segmentation tool will save the effort that is currently performed by radiologists in manual annotations for 70% of the cases. As the accuracy of the proposed work is >90% for 70% of the scans, the remaining 30% of scans, in which automatic segmentation fails, lack good preparation, such as (a) inadequate luminal distention, (b) imaging artifacts and distortion due to low-dose CT, (c) the patient is not accurately positioned, or (d) thickened colonic folds, as shown in Figure 9. For these cases, radiologists may use our annotation tool to manually refine the automatic results.

**Polyp Detection:** We want to emphasize that the current works of polyp detection focus on videos captured during real colonoscopy screening. So, the generated images are close to the FT virtual colonoscopy approach. Since we showed that FI has a better visualization performance than FT, we evaluate the proposed polyp detection approach on FI only, and we do not compare it with other approaches for a fair comparison.

In the first experiment, we generated videos using FI for 49 cases. Out of these 49 cases, there are 14 cases that have no polyps. The total number of polyps/lesions in the other 35 cases are 57 polyps (≥6 mm). We sampled these videos and obtained 10,600 frames with polyps, and 47,400 frames without polyps. Then, we used these frames to train our polyp detection model, which is RetinaNet [40] with ResNet-18 as a backbone. We used the Adam optmizer with a learning rate of $10^{-5}$, and for the focal loss, we used γ=2. Data were divided into six folds for the purpose of cross-validation. Figure 10 shows the Precision-Recall curve with an area under the PR curve of 0.93.

In the second experiment, we asked three radiologists to detect polyps for the 49 patients, using FI images (e.g., Figure 5). The radiologists did not have the ground truth of the polyps, and cases were ordered for each radiologist randomly to avoid any ordering effect. Sensitivity, specificity, and f1-score were calculated per patient; i.e., once a polyp is detected in a case and this patient has polyps, it counts as a true positive, no matter whether the other polyps are detected or not. In addition, if a polyp is detected in a healthy patient, it counts as a false negative. Table 3 shows the sensitivity, the specificity, and f1-score for the three radiologists (R #1, R #2, & R #3), as well as the proposed detector. The results show that the f1-score of our approach is higher than the radiologists’ scores. In addition, a 97% sensitivity confirms the low rate of the missing real polyps.

**CTC at low dose:** The results for the modified proposed algorithm are obtained by using 120 kVp (full dose) CT scans to create LDCTC at 80 kVp, 60 kVp, and 30 kVp. Using 30 KVp CTs, 70% of CTCs were successfully restored (i.e., the original information content was maintained), while at 80 kVp and 60 kVp CT 98%, a resemblance to the original CTC was obtained. To compare the accuracy of the construction, we segmented the reconstructed CTC scans to obtain the colon regions, to compare them with the segmented results from the standard dose CT. The comparison was performed by calculating the intersection over union for each CT slice, the average intersection over union, and the standard deviation was calculated for the whole dataset. Table 4 shows the results of this comparison. Using the Fly-In-based framework, the radiologists’ readings of the original and simulated LDCTC were as follows: at 30 Kvp, the sensitivity dropped to 92% relative to the standard dose, while at 60 Kvp, it dropped to 99.5% relative to the standard dose. This result can be further improved by training robust deep learning and enhancing the colon segmentation with respect to noise.

## 5. Conclusions and Future Work

In this work, we proposed a CTC-based framework for polyp detection. The proposed pipeline starts by segmenting the colon from its background. Then, the segmented colon is used to reconstruct the colon surface. Virtual cameras project the inner surface of the 3D colon, integrated by its curvature. These images were used to train a RetinaNet model for polyp detection. In addition, to test the effect of the variation of the CT dose on the proposed method, a prior denoising technique is used to remove the artifact resulting from low-dose CT scans. This technique showed that detecting polyps from enhanced images almost have the same performance as detecting polyps from standard dose CT. The experimental results confirm that the proposed approach can help gastroenterologists, and it holds a great promise for combating CRC.

Our future direction will focus on enhancing the segmentation algorithm using different deep learning techniques. Our goal is to correctly segment the colons from a CT scan that is not well prepped, and also the CT scans that are collected at very low doses. We will also focus on enhancing the automatic polyp detection module using a multistage polyp detection algorithm, which will be used in addition to our proposed polyp detection module in a second stage to directly detect the polyp from the original CT. Then, we use the correspondences between the candidate polyps in the 3D mesh and the CT slices to confirm the detected polyp and to reduce the false rate. Moreover, a polyp should be classified to identify its category.

## Figures and Tables

**Figure 1 sensors-22-09761-f001:**
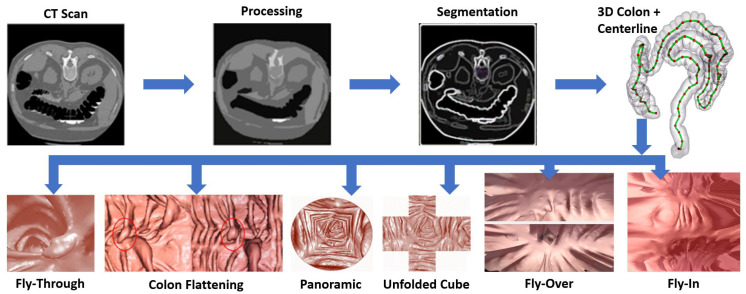
The CTC pipeline: First, DICOM images are cleaned, then colon regions are segmented. Secondly, the 3D colon is reconstructed from segmented regions, then centerline may be extracted. Finally, the internal surface of the colon can be visualized using different visualization methods.

**Figure 2 sensors-22-09761-f002:**
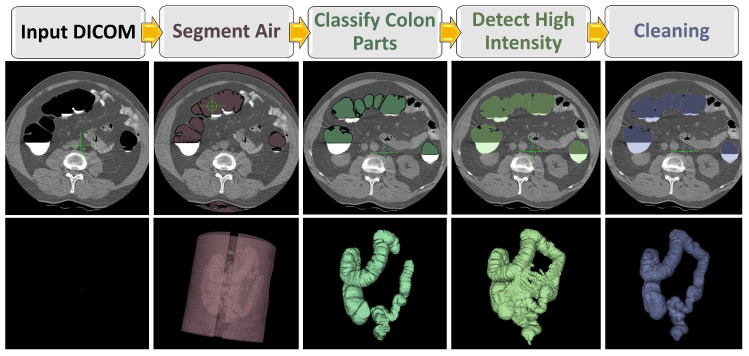
Automatic segmentation pipeline using an MRF model.

**Figure 3 sensors-22-09761-f003:**
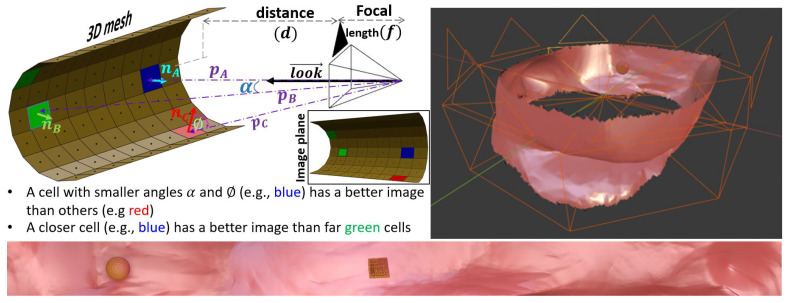
Visualization of colon segment via FI. (**Left**) Camera geometry showing that the cell visualization depends on: principal axis $\overset{⟶}{look}$, projection direction *p*, surface normal *n*, and distance *d*; (**right**) Cameras configuration of a rig of 8 cameras over a ring, and (**bottom**) the rendered filet.

**Figure 4 sensors-22-09761-f004:**
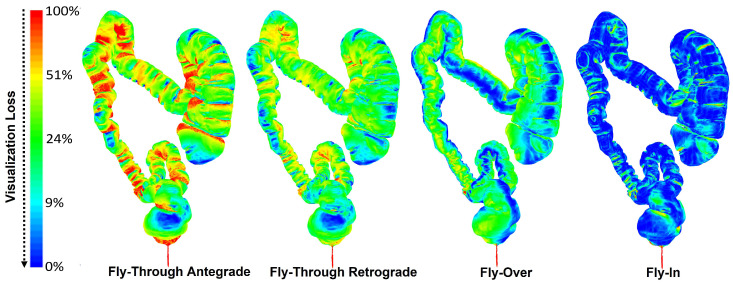
Graphical representation of the visualization loss using the image formation process in FT (both directions), FO, and FI.

**Figure 5 sensors-22-09761-f005:**
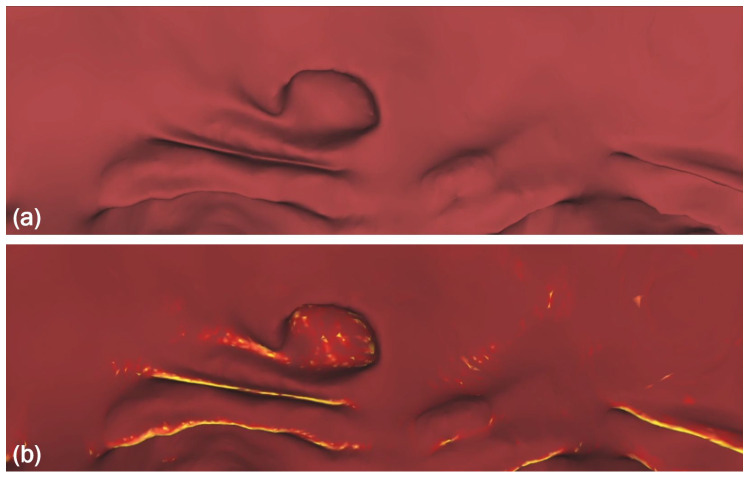
(**a**) An RGB image captured by virtual cameras. (**b**) Adding curvature information to the image highlights the convex and concave regions.

**Figure 6 sensors-22-09761-f006:**
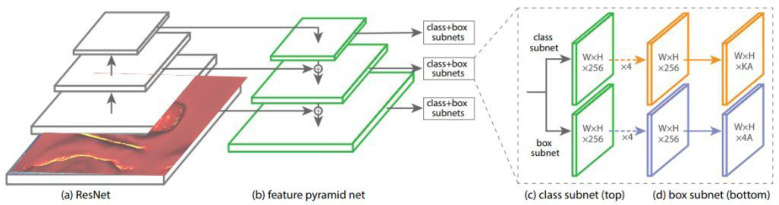
RetinaNet Model [40].

**Figure 7 sensors-22-09761-f007:**
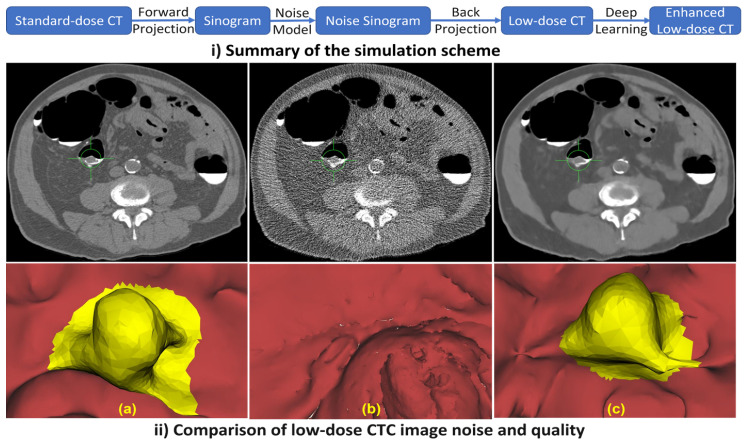
A patient with a 1.7 cm pedunculated polyp in the sigmoid colon. (**a**) Standard-dose CT is scanned (120 kVp/240 mAs). (**b**) Low-Dose CT is simulated using a noise insertion model (60 kVp). (**c**) Enhanced Low-Dose CT using FBP + U-Net back-projection approach [46,47].

**Figure 8 sensors-22-09761-f008:**
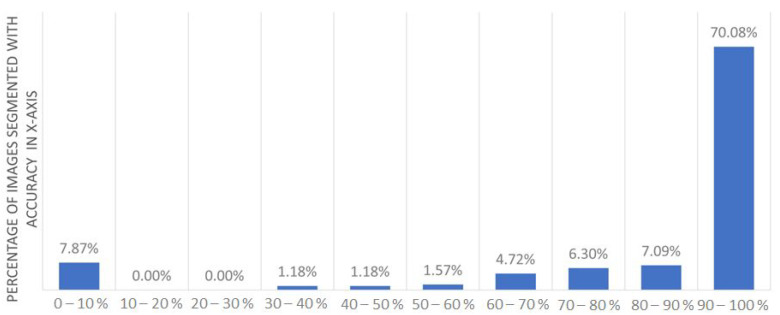
Segmentation accuracy is calculated using IoU metric for 254 DICOM scans.

**Figure 9 sensors-22-09761-f009:**
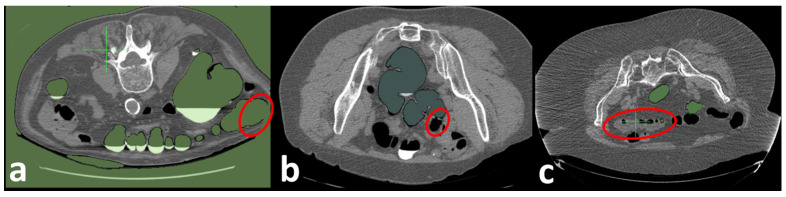
Examples of bad scans that need manual annotations: (**a**) Patient is not accurately positioned, (**b**) thickened colonic folds, and imaging artifacts and distortion due to low-dose CT.,(**c**) patient is not good prepped (colon is not empty).

**Figure 10 sensors-22-09761-f010:**
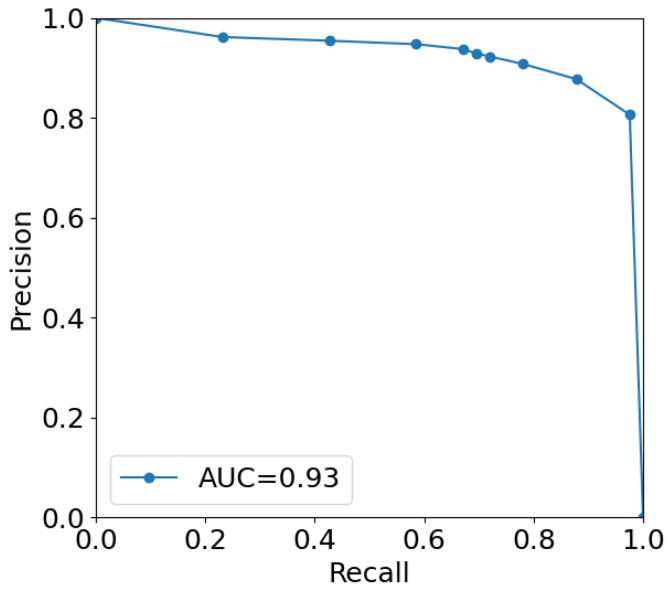
Precision-Recall curve for the cross-validation.

**Table 1 sensors-22-09761-t001:** Polyp detection approaches comparison.

Approach	Feature	Strength	Weakness
Multitarget stool DNA [27]	DNA	Non-invasive No need for special prep (empty colon) Low cost (∼$650 per scan) Widely available (sample can be collected at home)	Not real time (2 weeks) Cannot localize the polyp Low sensitivity in small polyp (6 mm) Not a treatment method
CT scans [28]	Shape, HU level	Non-invasive Obtains the polyp location Better sensitivity than multitarget stool DNA Real time detection Acceptable cost (∼$900 per scan)	Low sensitivity in small polyps (6 mm) Needs a special prep (empty colon) Not a treatment method
Optical colonoscopy [22,25,29,30]	Shape, Texture	Obtains the polyp location Best sensitivity Real time detection Could make a treatment while scanning No radiation exposure	Needs a special prep (empty colon) Invasive (can cause inflammation, bleeding, or perforation ) High cost (∼$2550 per scan) Requires sedation
Virtual colonoscopy (our approach)	Shape, Curveture	Non-invasive Obtains the polyp location High sensitivity Real time detection Acceptable costs same as CT (∼$900 per scan) No sedation	Need a special prep (empty colon) Not a treatment method Low dose of radiation

**Table 2 sensors-22-09761-t002:** Examples of automatic polyp detection algorithms.

Paper/Year	Modality	Model	Non-Invasive?	Sensitivity/ Specificity
Karkanis et al., 2003 [29]	OC videos	Color wavelet covariance + LDA	No	90%/97%
Godkhindi et al., 2017 [28]	CT scans	Three layer-CNN	Yes	88.8%/87.3%
Urban et al., 2018 [25]	OC videos	Preinitialized CNN	No	95%/96.8%
Zhang et al., 2019 [30]	OC videos	SSD-GPNet	No	76.4%/87.5%
Livovsky et al., 2021 [22]	OC videos	RetinaNet + LSTM-SSD	No	97.5%/95%

**Table 3 sensors-22-09761-t003:** Sensitivity, specificity, and f1-score for polyp detection.

	R #1	R #2	R #3	Ours
Sensitivity %	70	67	60	**97**
Specificity %	93	81	58	**79**
f1-score %	81	80	67	**94**

**Table 4 sensors-22-09761-t004:** Low-Dose reconstructed CT segmentation results comparison.

CT Tube Voltage	Percentage Reconstructed	IOU for Whole Dataset	IOU for Successfully Reconstructed Colon
30 KVp	92%	76.7%±0.07	94.5%±0.07
60 KVp	99.5%	97.4%±0.04	98.3%±0.04

## Data Availability

ACRIN Protocol 6664.

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
