# Peer review of "An AI-Based Colonic Polyp Classifier for Colorectal Cancer Screening Using Low-Dose Abdominal CT"

_sensors, 2022, doi:10.3390/s22249761_

Round 1
Reviewer 1 Report
I think it’s interesting research. I would recommend it if the following issues are handled properly.
1. Add more information about your datasets.
2. Add more details about your AI model of RetinaNet.
3. Why increased voltage could enhance the average relative sensitivities? Explain that. In addition, evaluate the voltage tolerance of the human body.
4. Highlight your novelties in the Abstract and conclusion.
5. Could you provide cartoon images and photos about the devices of a rig of 8 cameras over a ring in Figure 3?
6. The writings about this manuscript need to be improved. For example, “Developing accurate FI-based …… American College of Radiology.” This sentence should be ahead of the results in the Abstract.
Author Response
The authors are very thankful for the time and efforts by the honorable editor and referees for reviewing the manuscript and helping us to improve our research work.
Response to Comments and Suggestions from Reviewer 1
- Comment 1:
Add more information about your datasets.
Response: Thank you for helping us to improve our paper.
Action: More details are added.
Reference: Section: Experiments:dataset , Page 9.
- Comment 2:
Add more details about your AI model of Retina-Net.
Response: Thank you for helping us to improve the manuscript.
Action: More details are added.
Reference: Section: 2.2 Polyp Detection, Pages 6-7.
- Comment 3:
Why increased voltage could enhance the average relative sensitivities? Explain that. In addition, evaluate the voltage tolerance of the human body.
Response: The voltage of the tube of a CT scanner is one of the ways used to control the radiation level. To decrease the radiation levels, many CT scanners use low voltage which results in noisy reconstructed images.
The relative sensitivity reported in the result is the ratio between the sensitivity of the framework using reduced dose CT scans and using standard dose CT scans.
Action: A paragraph added to show the relation between the CT tube voltage, noise level and signal to noise ratio.
Reference: Section: 3. Low Dose CTC visualization using the FI approach, Page 8.
- Comment 4:
Highlight your novelties in the Abstract and conclusion.
Response: Thank you for helping us to improve our paper.
Action: Novelties are highlighted as advised.
Reference: Section: Abstract and Introduction, Pages 1,4.
- Comment 5:
Could you provide cartoon images and photos about the devices of a rig of 8 cameras over a ring in Figure 3.
Response: This rig of 8 cameras is a virtual device that is used on virtual colonoscopy not a real device. More detail could be found on references [18,31,33]. Fig 3. illustrates an example of a rig of 8 cameras. The figure shows their geometry, filed of views , ……
- Comment 6:
The writings about this manuscript need to be improved. For example, “Developing accurate FI-based …… American College of Radiology.” This sentence should be ahead of the results in the Abstract.
Response: Thank you for helping us to improve our manuscript.
Action: Review for language has been performed and the manuscript presentation is enhanced.
Reviewer 2 Report
In this manuscript, authors have proposed an AI-based Colonic Polyp Classifier for Colorectal Cancer Screening using Low Dose Abdominal CT. This manuscript is well written but requires some improvements, which are mentioned below.
1) Give some details of your mentioned contributions in the introduction section.
2) Related work is very short; please include more details related to it.
3) Include a comparison table in the related work that should highlight the strengths and weaknesses of the previous methods as well as the proposed method.
4) What is the motivation behind Retina-Net model selection for Polyp detection? Consider other models also and compare the results.
5) Can the deep learning approach be applied for colon segmentation instead of the MRF method? How will it affect the results?
Author Response
The authors are very thankful for the time and efforts by the honorable editor and referees for reviewing the manuscript and helping us to improve our research work.
Response to Comments and Suggestions from Reviewer 2
- Comment 1:
Give some details of your mentioned contributions in the introduction section
Response: Thank you for helping us to improve our paper.
Action: contributions are highlighted as advised.
Reference: Section: Introduction, Page 4 .
- Comment 2:
Related work is very short; please include more details related to it.
Response: Thank you for helping us to improve our paper.
Action: More detail for related work were added.
Reference: Section: Related Work and Table 1, Pages 3,4
- Comment 3:
Include a comparison table in the related work that should highlight the strengths and weaknesses of the previous methods as well as the proposed method.
Response: Thank you for helping us to improve our paper.
Action: Table added as advised.
Reference: Section: Table 1, Page 4 .
- Comment 4:
What is the motivation behind Retina-Net model selection for Polyp detection? Consider other models also and compare the results.
Response: Thank you for helping us to improve our paper.
Action: More detailed for Retina-Net were added including the motivation and reasons of using it. Considering other models will be performed in future works.
Reference: Section: 2.2. Polyp Detection, Page 6 .
- Comment 5:
Can the deep learning approach be applied for colon segmentation instead of the MRF method? How will it affect the results?.
Response: Thank you for helping us to improve our paper. AI and deep learning approach should improve the segmentation result, but it need a large dataset to be trained on.
Action: We already obtained a larger dataset and working on its annotation, in future work we will develop an AI approach for colon segmentation.
Reference: Section: 5. Conclusions and Future Work, Pages 11-12.
Reviewer 3 Report
This paper proposes an AI-Based Colonic Polyp Classifier for Colorectal Cancer
Screening using Low Dose Abdominal CT. Please take into account the following comments:
(1) Introduction is good and it is extended but due to that reason, we shouldn't reduce the importance of the "related work"/"literature review"/"preliminary" Sections. Do needful.
(2) Why are two Conclusion Sections (in Sec. 4 and Sec 5)in your work? This will misguide the readers.
(3) If the inclusion of Conclusion in Section 4 is a mistake then it may be Results but these results are insufficient.
(4) Future direction of the article is unclear.
(5) In the caption of Figure 7, why reference [37] is cited twice. Like, [37,38] [37]
Author Response
The authors are very thankful for the time and efforts by the honorable editor and reviwers for reviewing the manuscript and helping us to improve our research work.
Response to Comments and Suggestions from Reviewer 3
- Comment 1:
Introduction is good and it is extended but due to that reason, we shouldn't reduce the importance of the "related work"/"literature review"/"preliminary" Sections. Do needful.
Response: Thank you for helping us to improve our paper.
Action: More details on related work were added.
Reference: Section: Related Work and Table 1, Page 3,4 .
- Comment 2:
Why are two Conclusion Sections (in Sec. 4 and Sec 5)in your work? This will misguide the readers.
Response: Thank you for helping us to improve our paper. This was a mistake/typo.
Action: Word Conclusion was removed from Sec4 title.
Reference: Section: 4. Experiments, Page 9.
- Comment 3:
If the inclusion of Conclusion in Section 4 is a mistake, then it may be Results but these results are insufficient.
Response: Thank you for helping us to improve our paper.
Action: More detailed and results were added on the experiment section.
Reference: Section: 4. Experiments ,Fig 9 and Table 2 , Pages 9-11 .
- Comment 4:
Future direction of the article is unclear.
Response: Thank you for helping us to improve our paper.
Action: A paragraph for our future direction was added ad the end of the paper.
Reference: Section: 5. Conclusions and Future Work, Pages 11,12 .
- Comment 5:
In the caption of Figure 7, why reference [37] is cited twice. Like, [37,38] [37].
Response: Thank you for helping us to improve our paper. This was a mistake/typo.
Action: The additional citation was removed.
Reference: Section: Figure 7, Page 9.
Round 2
Reviewer 2 Report
I am not satisfied with your response to my comment 3. i.e.
Include a comparison table in the related work that should highlight the previous and proposed methods' strengths and weaknesses.
Author Response
The authors are very thankful for the time and efforts by the honorable editor and reviewers for reviewing the manuscript and helping us to improve our research work. We are very pleased that our response addressed most of your comments. We hope this round that our response addresses your suggestion in comment 3 properly.
- Comment 3:
“Include a comparison table in the related work that should highlight the strengths and weaknesses of the previous methods as well as the proposed method.”
Response: Thank you for helping us to improve our paper.
Action: We added an additional table to compare the different approaches for polyp detection. The comparison highlights the strengths and weaknesses of each approach. We also added a paragraph to emphasize the goal of this research.
“Table 1 compares between different approaches used for polyp detection. It illustrates the strengths and weakness of each category. Our goal in this work is to develop a non-invasive automatic polyp detection approach using virtual colonoscopy. Unlike optical colonoscopy, proposed approach can be used in a massive data screening. Also, it helps radiologists to accurately read abdominal scans in short time by detecting polyp candidates. This proposed approach could benefit many clinics especially ones in rural and economically depressed regions, which have limited number of gastroenterologists to do optical colonoscopy and experienced radiologist to read the scans.”
Reference: 1.2. Related Work: Table 1, Table 2. Pages 3-4.
Reviewer 3 Report
Authors addressed the earlier review comments
Author Response
The authors are very thankful for the time and efforts by the honorable editor and reviewers for reviewing the manuscript and helping us to improve our research work. We are very pleased that our response addressed your comments.